# Photoacoustic Tomography Appearance of Fat Necrosis: A First-in-Human Demonstration of Biochemical Signatures along with Histological Correlation

**DOI:** 10.3390/diagnostics12102456

**Published:** 2022-10-11

**Authors:** Yonggeng Goh, Ghayathri Balasundaram, Hui Min Tan, Thomas Choudary Putti, Celene Wei Qi Ng, Eric Fang, Renzhe Bi, Siau Wei Tang, Shaik Ahmad Buhari, Mikael Hartman, Ching Wan Chan, Yi Ting Lim, Malini Olivo, Swee Tian Quek

**Affiliations:** 1Department of Diagnostic Imaging, National University Hospital, 5 Lower Kent Ridge Road, Singapore 119074, Singapore; 2Translational Biophotonics Laboratory, Institute of Bioengineering & Bioimaging, Agency for Science, Technology & Research, 11 Biopolis Way, #02-02 Helios, Singapore 138667, Singapore; 3Department of Pathology, National University Hospital, 5 Lower Kent Ridge Road, Singapore 119074, Singapore; 4Department of Breast Surgery, National University Hospital, 5 Lower Kent Ridge Road, Singapore 119074, Singapore

**Keywords:** photoacoustic, lipoma, ultrasound, MRI

## Abstract

A 50-year-old woman with no past medical history presented with a left anterior chest wall mass that was clinically soft, mobile, and non-tender. A targeted ultrasound (US) showed findings suggestive of a lipoma. However, focal “mass-like” nodules seen within the inferior portion suggested malignant transformation of a lipomatous lesion called for cross sectional imaging, such as MRI or invasive biopsy or excision for histological confirmation. A T1-weighted image demonstrated a large lipoma that has a central fat-containing region surrounded by an irregular hypointense rim in the inferior portion, confirming the benignity of the lipoma. An ultrasound-guided photoacoustic imaging (PA) of the excised specimen to derive the biochemical distribution demonstrated the “mass-like” hypoechoic regions on US as fat-containing, suggestive of benignity of lesion, rather than fat-replacing suggestive of malignancy. The case showed the potential of PA as an adjunct to US in improving the diagnostic confidence in lesion characterization.

Fat necrosis, or cell death of adipose tissue, is a common benign condition that occurs from the lack of oxygen supply to adipose tissue [1]. As common causes include trauma or post-surgical changes [2], fat necrosis often presents as a palpable soft tissue mass at superficial regions [3].

Ultrasound (US) is the first-line imaging tool for these superficial lesions, but imaging appearances are extremely varied [4] due to the age of the lesion, which manifests as varying degrees of hardening, fibrosis, and degeneration. This often results in a diagnostic dilemma, which necessitates further cross-sectional imaging or invasive procedures, such as biopsy or excision for histological confirmation (Figure 1 and Figure 2). There is, hence, an unmet clinical need for an adjunct imaging modality to US to improve diagnostic capability.

Photoacoustic (PA) tomography, a hybrid optical imaging modality, is based on the light-induced ultrasound waves providing the contrast of optical imaging combined with the high spatial resolution of ultrasound [5]. Its ability to provide the distribution of endogenous chromophores, such as blood oxygenation [6], water [7], lipid [7,8,9,10] and recently collagen [11,12], makes it attractive as a potential adjunct tool to various aspects of ultrasound. This is particularly useful in regions abundant with these chromophores, such as superficial soft tissues and the breast, or in fat-containing and fibrotic/necrotic conditions, such as fat necrosis. However, these theoretical advantages and usages of chromophore differentiations have not been demonstrated in daily clinical usage. Herein, we present interesting images that demonstrate for the first time the biochemical signatures of fat necrosis derived by PA and its agreement with histopathology (Figure 3). A detailed description of PA imaging protocol and image reconstruction is included in Appendix A. This work showcases the potential of PA as an adjunct for US to improve the diagnostic confidence for fat necrosis.

In this case, the authors have successfully demonstrated biochemical features of fat necrosis on PA as a first in-human demonstration with histopathological correlation. PA was able to identify the “mass-like” hypoechoic regions on US as fat-containing, rather than fat-replacing. On pathology, the lipid signals on PA correspond to liquefied necrotic fat from cystic degeneration, while the collagen signals on PA correspond to the fibrosis around the cavity. Hence, the biochemical capability of fat and collagen characterization could help to resolve ambiguous findings on US and improve diagnostic confidence for fat necrosis. 

As it is crucial to obtain pathological correlation with US-PA images, excised tissues with no active blood signals had to be obtained. The authors believe the incorporation of blood signals in in vivo imaging would further intensify our understanding of the pathophysiology and respective imaging correlations for fat necrosis. Although the results from a single case may seem promising, more work must be done to validate these findings. In particular, more work must be done to validate findings of benign lipomas, fat necrosis, and malignant lipomatous tumors to investigate their biochemical differences. With more data, PA could potentially translate downstream into clinical imaging workflows for better characterization of superficial/breast lumps where fat-containing lesions are common. However, for widespread clinical adoption, there needs to be vast improvements in both hardware and software for PA imaging to improve its imaging depth (to at least 3–4 cm), its field of view, as well as spectral coloring.

## Figures and Tables

**Figure 1 diagnostics-12-02456-f001:**
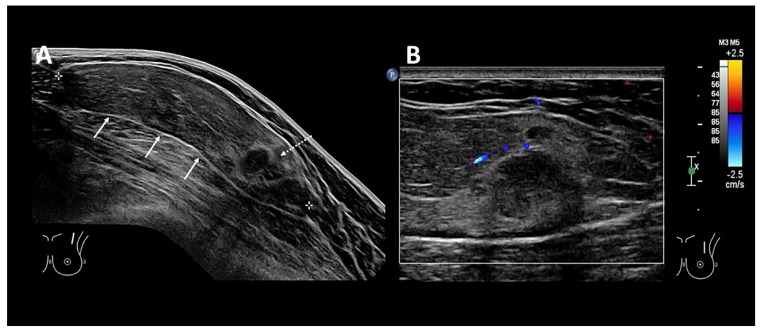
A 50-year-old woman with no past medical history presented with a left anterior chest wall mass. Clinically, the mass was soft, mobile, and non-tender. (**A**): US image of the left anterior chest wall mass shows a wider than tall mass with well-circumscribed margins (bold arrows). It contains internal fat echogenicity and is in keeping with a fat-containing lesion (i.e., lipoma). Within the inferior portions (dotted arow), there are focal “mass-like” hypoechoic nodules seen, which are surrounded by a hyperechoic capsule. (**B**): Doppler US of the focal “mass-like” nodules performed, demonstrated mild increased peripheral vascularity around the hyperechoic capsule but no increased vascularity within the hypoechoic “mass-like” nodules. Findings were indeterminate for malignant change of a lipomatous lesion.

**Figure 2 diagnostics-12-02456-f002:**
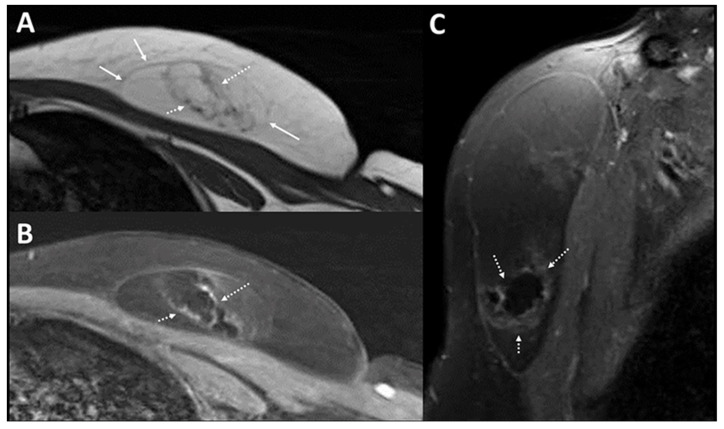
**MRI of left anterior chest wall.** MRI was performed for the patient in view of possible malignant change (**A**): Axial T1w image of the left anterior chest wall mass shows a fat-containing lesion in the left anterior chest wall (bold arrows) in keeping with a lipoma. Within the inferior portion, there are fat-containing areas on MRI, which are surrounded by T1w hypointense irregular bands (dotted arrows). (**B**): These irregular T1w hypointense bands show enhancement on the post contrast enhanced image. (**C**): Sagittal post contrast enhanced image shows similar findings of irregular rim enhancement around a focal fat-containing central component within inferior portions of the lipoma.

**Figure 3 diagnostics-12-02456-f003:**
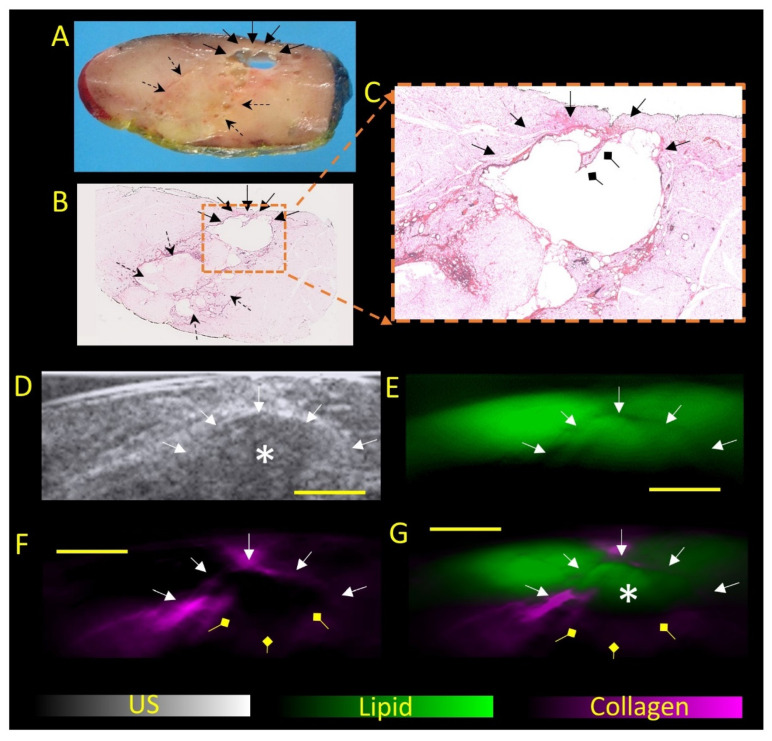
PA imaging of the excised tissue. Patient underwent uneventful excision of the left chest wall mass and was imaged with PA. Methods are as described in Appendix A. (**A**): Gross pathology of a representative cut section of the lipomatous tumor showing yellowish fatty appearance and foci of necrosis (dotted arrows). Focal cystic change containing oily fluid secondary to fat necrosis was present at where the “mass-like” nodules were seen on US (bold arrows). This area is rimmed by fibrosis. (**B**): H&E-stained microscopic image of the lipomatous tumor confirms prominent degenerative change, featuring areas of fat necrosis (dotted arrows), cystic change (bold arrows), and fibrosis. (**C**): shows zoomed in image of the cyst with septation (diamond arrow). This largest area of cystic change (bold arrows) corresponds to the “mass-like” nodule as seen on ultrasound (labelled as *). (**D**): The “mass-like” nodule near the posterior margin (*), surrounded by a hyperechoic rim (bold arrow), was targeted for PA imaging. Corresponding PA images showing distribution of lipid (**E**) and collagen (**F**), or their overlay (**G**) demonstrated collagenous signal corresponding to the hyperechoic halo in keeping with fibrosis. Within the “mass-like” region, the imaged portions demonstrated lipid signal, which was similar in intensity to the surrounding lipoma. No suspicious fat or collagenous-replacing masses were identified. No blood signals were obtained from this ex vivo study as there was no active ongoing blood flow after lesion excision. Scale bar 5 mm.

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
