# Peer review of "Photoacoustic Tomography Appearance of Fat Necrosis: A First-in-Human Demonstration of Biochemical Signatures along with Histological Correlation"

_diagnostics, 2022, doi:10.3390/diagnostics12102456_

Round 1
Reviewer 1 Report
This paper suggests that photoacoustic tomography can be used as an auxiliary imaging method in clinical practice to improve the diagnostic ability of ultrasound imaging. In a real case, photoacoustic imaging of endogenous chromophore such as lipid and collagen was used to demonstrate the biochemical characteristics of fat necrosis. This is the first time that the correlation between biochemical characteristics and histology has been demonstrated in humans, demonstrating the potential of PA as an aid to ultrasound imaging and improving the diagnostic confidence of pathological features.
However, the similar results have been reported by Photoacoustics in September 2022, the title is “Biochemical “decoding” of breast ultrasound images with optoacoustic tomography fusion: First-in-human display of lipid and collagen signals on breast ultrasound”, so a major modification is suggested for acceptance. The main recommendations are as follows:
(1) Since similar results have been reported, the innovation point of this article needs further refinement.
(2) It is suggested that the anatomical orientation of the ultrasound images in Figure 1 be improved to enhance comprehensibility.
(3) It is recommended that the authors briefly introduce the absorption spectral properties of collagen and lipids, and explain the reasons for the wavelength used in the Materials and Methods section.
(4) Figure 3 has slightly less photoacoustic images and insufficient evidence. If possible, it is suggested to add photoacoustic images such as XZ,YZ orientation.
Reviewer 2 Report
In this manuscript, Goh et al. reported the photoacoustic imaging (PAI) of an excised tumor specimen for biochemical analysis and suggested the potential usage of PAI for improved breast cancer diagnosis. The authors are suggested to mention the challenges of translating ex vivo imaging into in vivo imaging, including imaging depth, limited view, and spectral coloring.
Round 2
Reviewer 1 Report
The revised paper provides a reasonable explanation of the issues raised. The experimental results show that photoacoustic imaging can improve the diagnostic reliability of fat necrosis, which has potential practical significance in clinical practice.
The authors may consider including the current status of research on lipid composition by photoacoustic tomography in the reference section, such as Characterization of lipid-rich aortic plaques by intravascular photoacoustic tomography: ex vivo and in vivo validation in a rabbit atherosclerosis model with histologic correlation in the Journal of the American College of Cardiology and Reliability assessment on intravascular photoacoustic imaging of lipid: ex vivo animal and human sample validation in the Journal of Biophotonics.
Author Response
Dear Reviewer,
Many thanks for the suggestion of references. The references have been added into the manuscript and the sentence has been slightly edited to incorporate these references as suggested. The sentence now reads as:
"Its ability to provide the distribution of endogenous chromophores like blood oxygenation (6), water (7), lipid (7-10) and recently collagen (11, 12) makes it attractive as a potential adjunct tool to various aspects of ultrasound."